# Investigation of Mating Pheromone–Pheromone Receptor Specificity in *Lentinula edodes*

**DOI:** 10.3390/genes11050506

**Published:** 2020-05-04

**Authors:** Sinil Kim, Byeongsuk Ha, Minseek Kim, Hyeon-Su Ro

**Affiliations:** Division of Applied Life Science and Research Institute of Life Sciences, Gyeongsang National University, Jinju 52828, Korea; zerolune@naver.com (S.K.); h89a@naver.com (B.H.); kmstaur@gmail.com (M.K.)

**Keywords:** *Lentinula edodes*, mating pheromone, mating receptor, specificity

## Abstract

The *B* mating-type locus of *Lentinula edodes*, a representative edible mushroom, is highly complex because of allelic variations in the mating pheromone receptors (*RCB*s) and the mating pheromones (*PHB*s) in both the *Bα* and *Bβ* subloci. The complexity of the *B* mating-type locus, five *Bα* subloci with five alleles of *RCB1* and nine *PHBs* and three *Bβ* subloci with 3 alleles of *RCB2* and five *PHB*s, has led us to investigate the specificity of the PHB–RCB interaction because the interaction plays a key role in non-self-recognition. In this study, the specificities of PHBs to RCB1-2 and RCB1-4 from the *Bα* sublocus and RCB2-1 from the *Bb* sublocus were investigated using recombinant yeast strains generated by replacing *STE2*, an endogenous yeast mating pheromone receptor, with the *L. edodes*
*RCB*s. Fourteen synthetic PHBs with C-terminal carboxymethylation but without farnesylation were added to the recombinant yeast cells and the PHB–RCB interaction was monitored by the expression of the *FUS1* gene—a downstream gene of the yeast mating signal pathway. RCB1-2 (*Bα2*) was activated by PHB1 (4.3-fold) and PHB2 (2.1-fold) from the *Bα1* sublocus and RCB1-4 (*Bα4*) was activated by PHB5 (3.0-fold) and PHB6 (2.7-fold) from the *Bα2* sublocus and PHB13 (3.0-fold) from the *Bα5* sublocus. In particular, PHB3 from *Bβ2* and PHB9 from *Bβ3* showed strong activation of RCB2-1 of the *Bβ1* sublocus by 59-fold. The RCB–PHB interactions were confirmed in the monokaryotic S1–10 strain of *L. edodes* by showing increased expression of *clp1,* a downstream gene of the mating signal pathway and the occurrence of clamp connections after the treatment of PHBs. These results indicate that a single PHB can interact with a non-self RCB in a sublocus-specific manner for the activation of the mating pheromone signal pathways in *L. edodes*.

## 1. Introduction

The mating process of mushrooms initiates when the mycelial hyphae of compatible monokaryotic strains fuse together to generate new dikaryotic hypha. Compatibility of the mating is determined by genes lying at the chromosomal *A* and *B* mating-type loci, which are involved in various steps of the mating process, including hyphal fusion, nuclear pairing, clamp cell formation and nuclear division, clamp cell fusion and nuclear migration to establish dikaryons throughout the connected mycelial cells [1,2,3]. The *B* mating-type locus has been of particular interest because it is composed of mating pheromone genes (*PHB*s) and pheromone receptor genes (*RCB*s) that play a key role in non-self-recognition at the first stage of the whole mating process. *RCB*s encode G-protein-coupled receptors (GPCRs), which consist of seven transmembrane domains, an extracellular mating pheromone-interacting domain an intracellular domain that interacts with the α subunit of heterotrimeric G protein [3]. *PHB*s encode pre-pheromone polypeptides of 53–72 amino acids as shown in *Coprinopsis cinerea* and *Schizophyllum commune* [4,5,6]. The pre-pheromones undergo a maturation process to form a mature pheromone which includes proteolytic cleavage of the N-terminal pre-peptide, removal of three amino acids (AAX) at the C-terminal CAAX motif (C, cysteine; A, aliphatic residue; X, any residue), carboxymethylation of the carboxyl group and farnesylation of the sulfhydryl group at the terminal cysteine residue [7,8,9]. The whole maturation process yields mature lipopeptide pheromones with 11–13 amino acid residues [5,10]. Mushroom mating pheromones and pheromone receptors are homologous to the non-diffusible mating pheromone a-factor and the pheromone receptor Ste3 of *Saccharomyces cerevisiae* [1,3,11,12].

The mating pheromone signal pathway in *S. cerevisiae* is initiated by the interaction between the mating pheromone and pheromone receptor, which results in the activation of Gpa1, the α subunit of the heterotrimeric G-protein complex. The activated GTP-loaded Gpa1 dissociates from the G-protein αβγ complex [13,14], and then the freed G-βγ complex transfers the mating signal to the nucleus through the mitogen-activated protein kinase (MAPK) cascade to activate Ste12, which regulates the expression of mating response genes such as *FUS1*, *FUS2*, *SST2*, *FIG1* and *FIG2* [14,15]. Activation of the mating pheromone pathway results in the cell cycle arrest at the G1 phase and the morphologic change to the “shmoo” form [16]. Fungi belonging to Basidiomycota are known to contain a similar mating pheromone signal pathway to that of yeast [1]. Physical interaction between the mating pheromone and the mating pheromone receptor initiates the mating pheromone signal pathway to activate Mat2, a transcription factor that positively regulates the mating-related genes, including *znf2*, *HD* genes of the *A* mating-type locus and *clp1* [3,17,18,19,20,21,22,23]. The *clp1* gene is involved in the formation of clamp cells; therefore, the elevated expression of *clp1* in the mated dikaryotic mycelia has been considered as a hallmark of successful mating in most Basidiomycota [24,25].

Unlike *S. cerevisiae*, the mushroom *B* mating-type locus is highly complex. The *B* mating-type locus of *C. cinerea* consists of three subloci, each of which carries one *RCB* and two *PHB*s. Moreover, *RCB*s are found to carry four or five different allelic variations, resulting in 79 *B* mating types [1,4,23]. *S. commune* also has similar multiple subloci and allelic variations within the subloci, comprising 81 *B* mating types [26]. Similar to *S. commune*, the *B* mating-type locus of *Lentinula edodes* has two subloci, *Bα* and *Bβ*, and each sublocus contains one *RCB* and two *PHB*s, with one exception [27]. In the previous study, we revealed that there are five *Bα* subloci with five alleles of *RCB1* and nine *PHB*s and three *Bβ* subloci by three alleles of *RCB2* and five *PHB*s [27]. However, the specificity between the 14 mating pheromones and the eight alleles of pheromone receptors are yet to be verified. To this end, we constructed yeast model systems that express mushroom mating pheromone receptors by replacing yeast *STE3* with *L. edodes RCB*s. Interaction of PHB and RCB was monitored through the expression of *FUS1,* a mating response gene whose expression is increased by pheromone signals in the yeast cells [28]. In this study, synthetic PHBs with the C-terminal carboxymethylation, but without farnesylation, which were proven to be functionally active in the mating pathway activation in *L. edodes* [12], were applied in order to study the PHB–RCB interactions.

## 2. Materials and Methods

### 2.1. Strains and Culture Conditions

*S. cerevisiae* W303a strain (*MATa, ade2, can1-100, his3-11,15, leu2-,112, trp1-1, ura3-1*) was used to study PHB–RCB interaction of *L. edodes*. *S. cerevisiae* was cultured at 30 °C in yeast extract-peptone-dextrose medium (YPD; 1% yeast extract, 2% peptone, 2% glucose) or synthetic defined medium (SD; 0.17% yeast nitrogen base, 0.5% ammonium sulfate, 2% glucose) with amino acid supplementation. *L. edodes* was grown on potato dextrose broth (PDB; Difco, Sparks, MD, USA) or potato dextrose agar (PDA; Difco) at 25 °C.

### 2.2. Modification of Yeast Host Strain

Two modifications were introduced to *S. cerevisiae* to enhance the mating pheromone signal from the PHB–RCB interaction. First, *SST2* was deleted by transformation of a deletion module (Promoter *SST2*–*HIS3*–Terminator *SST2*) and subsequent selection against SD agar without histidine to generate the strain RCY1420 (Table 1, Appendix A). Second, the C-terminal amino acids, K-I-G-G-I, of yeast Gpa1 were replaced with E-A-G-L-L of *L. edodes* Gpa1 homolog. For this, the strain RCY1432 was generated by transformation with the modification module (C-*GPA1-EAGLL* (250 bp)–*TRP1*– Terminator of *GPA1*) to the strain RCY1420 and subsequent selection against SD agar without histidine and tryptophan (Table 1, Appendix A). Transformation of the yeast cell was performed by the PEG-lithium acetate method using salmon sperm DNA (Invitrogen, Carlsbad, CA, USA) as a carrier DNA. The *HIS* and *TRP* markers were from the yeast 2-micron vectors pRS423 and pRS424, respectively.

### 2.3. Total RNA Extraction and PCR Conditions

*RCB* genes in *L. edodes* were obtained by reverse transcriptase-PCR (RT-PCR) using *L. edodes* total RNA as a template. For this, total RNA was extracted from mycelial powder of *L. edodes* grown on PDA for 14 days using RNeasy Plant Mini Kit (Qiagen, Hilden, Germany). The extracted total RNA was subjected to RT-PCR to obtain cDNA using a HiPi RT-Prime Kit (ELPIS, Daejeon, Korea). PCR was performed with the cDNA and the primer sets specific to *RCB* genes (Appendix A) under the following conditions: hold for 10 min at 95 °C; 25 cycles of 95 °C for 30 s and 60 °C for 30 s and 72 °C for 30–120 s; and 72 °C for 5 min. PCR products were analyzed by 1% agarose gel electrophoresis and recovered using a Gel Elution Kit (Geneall, Seoul, Korea).

### 2.4. Construction of RCB-Expressing Yeast Strains

*STE2* of *S. cerevisiae* was replaced with *L. edodes RCB*s. The replacement modules were generated as *STE2* promoter (P*_STE2_*, 348 bp)-*RCB-KanMX6*-*STE2* terminator (T*_STE2_*, 399 bp). The transformants were selected against YPD-geneticin (100 μg/mL) (Sigma-Aldrich, St. Louis, MO, USA) using *KanMX6* gene from pFA6a-*KanMX6* vector as a selective marker (Appendix A). Yeast strains constructed for this study are summarized in Table 1.

### 2.5. Pheromone Response Assay with Synthetic PHBs

Pheromone peptides containing methyl ester at C-terminal cysteine residue were synthesized through a commercial service (GenScript, Piscataway, NJ, USA) (Table 2). The synthetic PHBs (20 μg/mL) were applied to the recombinant yeast culture grown in fresh YPD (OD600 = 0.6). After incubation for 6 h at 25 °C, cells were harvested by centrifugation (3500 rpm, 10 min) and were washed twice with distilled water. The recovered cells were subjected to total RNA extraction using TRI-RNA reagent (Invitrogen). The expression of *FUS1* as a marker of PHB–RCB interaction in yeast was monitored through real-time qRT-PCR using TOPreal™ qPCR 2X PreMix (Enzynomics, Daejeon, Korea). Real-time qPCR was performed under the following conditions: Hold for 10 min at 95 °C; 45 cycles of 95 °C for 20 s, 57 °C for 30 s and 72 °C for 20 s. Relative expression of the target gene was calculated based on the Cq value of *β-tubulin* gene. All data were obtained in triplicates from three independent experiments. Statistical significance of the mean difference was examined by one-way ANOVA using Microsoft Excel program with Real Statistics Resource Pack software (www.real-statistics.com).

For the flow cytometry analysis, the yeast cells grown at the exponential growth phase (OD600 = 0.2) were harvested. The collected cells were suspended in fresh YPD medium containing 20 μg/mL of PHB and then incubated for 3 h at 25 °C. After the incubation, the yeast cells were collected and then washed twice with phosphate-buffered saline (PBS). The washed cells (2 × 10^6^ cells) were resuspended in 1 mL of cold PBS. A volume of 9 mL of 70% ethanol was added to the cell suspension with gentle shaking. The suspension was incubated at −20 °C for 12 h. For the propidium iodide (PI, Sigma-Aldrich) staining, the supernatant was removed after centrifugation and the collected cells were washed with PBS. The PBS-suspended cells were treated with RNase A (100 μg/mL) for 2 h at 37 °C. PI stock solution was added to achieve the final concentration of 20 μg/mL. The PI-treated cells were subjected to flow cytometry analysis using BD FACSVerse^TM^ (Becton Dickinson).

### 2.6. Effect of Synthetic PHBs on Lentinula Edodes S1–10

The monokaryotic S1–10 strain of *Lentinula edodes* was grown in PDB (100 mL) for a week at 25 °C. The mycelia were harvested by centrifugation. The collected mycelia were suspended in fresh PDB and were fragmented by shaking (150 rpm) in the presence of 5 pieces of glass bead (*d* = 1 cm, Sigmund Linder GmbH, Germany) for 12 h. The culture broth was incubated additional 2 days after removal of the glass beads. The mycelia were washed twice with fresh PDB after centrifugation. The mycelia were resuspended in 100 mL PDB. Synthetic PHB (final 20 μg/mL) was added to the mycelial suspension at 25 °C. Samples (10 mL) were taken at given time intervals and were subjected to total RNA extraction. The real-time qRT-PCR analysis was performed using the same procedure described above with primer sets specific to target genes (Appendix A).

For the microscopic analysis, the PHB-treated mycelia were stained with equivalent mixture of 10% potassium hydroxide and calcofluor white (Sigma-Aldrich), which binds to the chitin component of the fungal cell wall. The stained mycelia were examined under fluorescence microscope (DP-70, Olympus, Japan) at 1000X magnification with excitation at 355 nm.

## 3. Results

### 3.1. Construction of Yeast Model System for the Investigation of Lentinula edodes Mating Pheromone Receptor–Pheromone Interactions

The yeast strain RCY1419 carrying *L. edodes RCB1-4* instead of yeast *STE2* did not respond to the treatment of *L. edodes* mating pheromones, PHB5 and PHB6, which had been effective in the activation of RCB1-4 in *L. edodes* [12] (Figure 1a). To sensitize the *L. edodes* RCB to the mating pheromone in yeast, *SST2* in the host yeast cell was deleted. Sst2 is a negative regulator of the mating pheromone signal pathway [29] and its deletion is known to sensitize the mating pheromone signal in yeast [30]. However, RCB1-4 in *sst2Δ* background (RCY1421) was still unresponsive to PHB5 and PHB6 (Figure 1a). We next replaced the C-terminal five amino acids (KIGGI) of the yeast Gpa1 with the corresponding C-terminal five amino acids (EAGLL) of *L. edodes* Gpa1. The C-terminal region of Gpa1 is important for the receptor–Gpa1 interaction and the C-terminal modification can increase the interaction between heterogenous receptors in *S. cerevisiae* [31]. These modifications resulted in the stable response of RCB1-4 to PHB treatment by showing a 5.3-fold increase in *FUS1* expression (Figure 1a), even though the RCB1-4-PHB5/PHB6 interaction was not fully comparable to the Ste2–α-factor interaction, which showed a 24-fold increase in the *FUS1* expression and complete change in the yeast morphology (Appendix A). The optimal PHB treatment time was 6 h, and the *FUS1* level decreased beyond 6 h (Figure 1b). The *FUS1* expression increased along with the increase in PHB concentration. However, it gradually decreased after the peak at 20 μg/mL (Figure 1c).

### 3.2. Interaction between Lentinula edodes RCBs and PHBs in the Yeast Model System

Yeast strains carrying eight *RCB*s were constructed using RCY1432 as a background strain. Significant activation of the yeast mating pheromone signal pathway was observed from the yeast strains carrying *RCB1-2*, *RCB1-4* and *RCB2-1* after the treatment of each of the 14 PHBs at the concentration of 20 μg/mL whereas other *RCB*s were not responsive to the PHBs for unknown reason (Appendix A). We focused on these three *RCB*s for further investigation.

The mating pheromone receptor *RCB1-2* is one of the alleles of *RCB1* found from the *Bα* sublocus of the *B* mating-type locus in the commercial strains of *L. edodes*, including SJ701, SJ707, KFRI619, SMR1 and SMR2 [27]. It has two associated mating pheromone genes, *PHB5* and *PHB6* [27]. To investigate the interaction between RCB1-2 with 14 PHBs found from different alleles of the 5 *Bα* and 3 *Bβ* subloci (Figure 2a), the synthetic PHBs were added with the yeast RCY1433 strain, which was created by incorporating *RCB1-2* into the RCY1432 strain. In the single PHB treatment, only PHB1 and PHB2 showed a 4.3-fold and 2.1-fold increase in the *FUS1* expression, respectively, while there was no response by other PHBs, including the self PHBs (PHB5 and PHB6) (Figure 2b). Treatment of PHBs in the pair showed a 2.7-fold increase in *FUS1* expression only from the PHB1/PHB2 pair, which constitutes the Bα1 sublocus with the mating pheromone receptor *RCB1-1* (Figure 2c). This indicates that a single PHB is enough and that PHBs in a pair are not essential for activation of the mating pathway. It also suggests that *RCB1-2* in *Bα2* interacts with PHBs from the *Bα1* sublocus to activate the mating pathway in *L. edodes*.

We next examined the effects of PHB on *RCB1-4* from the *Bα* sublocus of *L. edodes* and *RCB2-1* from the *Bβ* sublocus using the yeast RCY1434 and RCY1440 strains, respectively. The enhanced *FUS1* expression through the interaction with RCB1-4 from PHB5, PHB6 and PBH13 was found to be 3.0-, 2.7- and 3.0-fold, respectively (Figure 3a). The interaction of PHB5 and PHB6 from the *B2* sublocus with RCB1-4 is already known in *L. edodes* [12]; however, the activation by PHB13 from the *Bα5* sublocus with RCB1-4 is a new finding. Treatment of PHBs in a pair did not show synergistic effect by PHB13/PHB14 from the *Bα5* sublocus, however, PHB5/PHB6 from the *Bα2* sublocus showed additive effect in the interaction with RCB1-4 (Figure 3b). RCB2-1 from the *Bβ1* sublocus was highly activated by PHB3 and PHB9 from *Bβ2* and *Bβ3* subloci, respectively (Figure 3c). PHB3– or PHB9–RCB2-1 interaction resulted in a 59-fold increase in the *FUS1* expression, which was even higher than the activation by the Ste2–α-factor interaction (24-fold). However, the treatment of both pheromones from *Bβ2* (PHB3 and PHB4) or *Bβ3* (PHB9 and PHB10) was rather inferior to the individual treatment of PHB3 or PHB9 (Figure 3d), suggesting that the presence of nonspecific pheromone inhibits the specific PHB–RCB interaction.

In yeast, the mating pathway activation by Ste2–α-factor interaction results in cell cycle arrest at the G1 phase accompanied by a morphologic change to the “Shmoo” form (Figure 4). We analyzed the effect of the RCB–PHB interaction on the yeast cell cycle and cell morphology. The RCB2-1–PHB9 interaction, which showed strong activation of the *FUS1* expression, resulted in complete morphologic change and cell cycle arrest (Figure 4a,b), whereas, regarding the effects of RCB1-2–PHB1 or RCB1-4–PHB5, PHB6 interactions were less severe. In these cases, although the interactions appeared to fail to arrest the cell division cycle, both interactions resulted in swollen cell morphologies, but not the “Shmoo” form (Figure 4b).

### 3.3. Effects of PHBs on the Activation of the Lentinula edodes Mating Pathway

The effect of PHB on the mating pathway activation in *L. edodes* was investigated by the treatment of PHB1 or PHB9 to the monokaryotic *L. edodes* S1–10 strain, whose *B* mating-type locus consists of *Bα2* (*RCB1-2*) and *Bβ1* (*RCB2-1*) subloci. Although it would be ideal to use knock-out strain that only carries single *RCB*, our investigation was carried out using the S1–10 strain without modification due to lack of available transformation method for *L. edodes*. It is notable that the sublocus specific PHB–RCB interaction seen in the yeast model study may exclude cross-activation by PHB1–RCB2-1 or PHB9–RCB1-2 interaction. 

Upon treatment of PHB1 or PHB9, there were small changes in the expression of *HD1*, *HD2* and *znf2* (Figure 5). A slight decrease in *HD1* and *HD2* expression was observed 12 h after the PHB1 treatment whereas the expression of *znf2* was gradually decreased (Figure 5). On the other hand, the transcript levels of *HD1* and *HD2* were peaked after 6–12 h after the PHB9 treatment while the level of *znf2* was unchanged. However, *clp1*, a downstream gene of *HD1* and *HD2*, showed a strong gradual increase in gene expression during 24 h of incubation (Figure 5). This trend was also true for the PHB9 with more *clp1* expression. The elevated expression of *clp1* has been considered as a hallmark of successful mating in mushrooms [24,25]. The expression of *priA*, which may play a role during the beginning of fruiting body formation, increased shortly after the treatment of PHB1 or PHB9. However, it decreased gradually 3–6 h later (Figure 5). The effect of PHBs was further confirmed by the occurrence of clamp connection, as evidence of the mating pathway activation in the monokaryotic strain (Figure 5).

## 4. Discussion

Mating of fungi belonging to Basidiomycota is initiated by the interaction between pheromone (PHB) and pheromone receptor (RCB). PHBs and RCBs are, necessarily, multiallelic to increase mating opportunities in the wild environment since the specificity of the PHB–RCB interaction is an important factor in self/non-self-recognition in the mating process [1,4,11,31,32]. In accordance with this, *C. cinerea* [1,4,23] and *S. commune* [26] are known to have more than 79 and 81 *B* mating types, respectively. The PHB–RCB interaction in *C. cinerea* occurs selectively between non-self PHB and RCB [10,14,23]. Studies using *S. cerevisiae* as a model system have shown that the fungal *RCB*s expressed in the yeast cells interact selectively with synthetic PHBs or plasmid-expressed PHBs from *C. cinerea* [18,33] and *S. commune* [26,34]. The *B* mating-type of *L. edodes* is known to consist of five *Bα* subloci by the combination of five alleles of *RCB1* and nine *PHB*s and three *Bβ* subloci by combination of three alleles of *RCB2* and five *PHB*s [27,35]. Due to the complexity of PHBs and RCBs, it was necessary to investigate the interaction of PHB and RCB in *L. edodes*. As in the study in *C. cinerea* [18,33] and *S. commune* [26,34], we have used synthetic PHBs and the RCB-expressing yeast strain.

In our yeast model, the C-terminal modification of Gpa1 and the deletion of *SST2* were necessary to sensitize the RCB interaction with Gpa1 (Figure 1)—a mediator of pheromone signal to the mating pheromone signal pathway—as previously reported [30,31]. In the treatment of synthetic PHBs of the RCB-expressing yeast strain with modified *GPA1* and *SST2* deletion, RCB1-2 from the *Bα2* sublocus showed a moderate interaction with PHB1 and PHB2 from the *Bα1* sublocus (Figure 2). RCB1-4 from the *Bα4* sublocus also showed moderate interactions with PHBs (PHB5 and PHB6) from the *Bα2* sublocus and PHB13 from the *Bα5* sublocus (Figure 3). RCB2-1 from the *Bβ1* sublocus showed a strong interaction with PHB3 from the *Bβ2* sublocus and PHB9 from the *Bβ3* sublocus, which was even higher than that of the Ste2–α-factor interaction in *S. cerevisiae* (Figure 3). The specific PHB–RCB interaction revealed from the yeast study was further confirmed by the expression of *clp1* and the formation of a clamp connection in the monokaryotic *L. edodes* S1–10 strain, as indications of the mating pheromone pathway activation.

It appears that RCBs in *L. edodes* have a specificity toward a certain group of PHBs. In our previous study, we categorized the *L. edodes* PHBs into three different groups based on the amino acid composition: PHB1, PHB2, PHB8 and PHB14 to group I; PHB5, PHB6, PHB11 and PHB13 to group II; and PHB3, PHB9, PHB7 and PHB10 to group III [27]. RCB1-2 in this study shows interaction with the group I PHBs (PHB1 and PHB2) whereas RCB1-4 and RCB2-1 interact with the group II PHBs (PHB5, PHB6 and PHB13) and group III PHBs (PHB3 and PHB9), respectively. The results imply that the membrane-embedded RCBs have group-specific recognition motifs in their loop domains, potentially in the L3, L5 and L7 domains described in our previous report [27]. It also implies that the non-self recognition in the mushroom mating can occur though the PHB group-specific interaction with RCBs.

C-terminal farnesylation in fungal PHBs is known to be important for the function of pheromone [36,37,38,39,40]. Little is known about the maturation of mating pheromones in *L. edodes*. However, the presence of CAAX motif suggests possible farnesylation at the C-terminus. The present study demonstrates that some synthetic PHBs can activate the mating pheromone signal pathway without the C-terminal farnesylation. However, the C-terminal farnesylation appears to be requisite for the full function of PHB since the synthetic PHBs without farnesylation failed to interact with other *L. edodes RCB*s in the yeast model (Appendix A).

## 5. Conclusions

Through the yeast model study, we discovered that a single PHB, despite found as a pair to constitute a sublocus, is enough to activate a certain RCB. We also found that RCB is only activated by PHB of the same sublocus; that is, the *Bα* receptors are specific to *Bα* PHBs and *Bβ* receptors are specific to *Bβ* PHBs.

## Figures and Tables

**Figure 1 genes-11-00506-f001:**
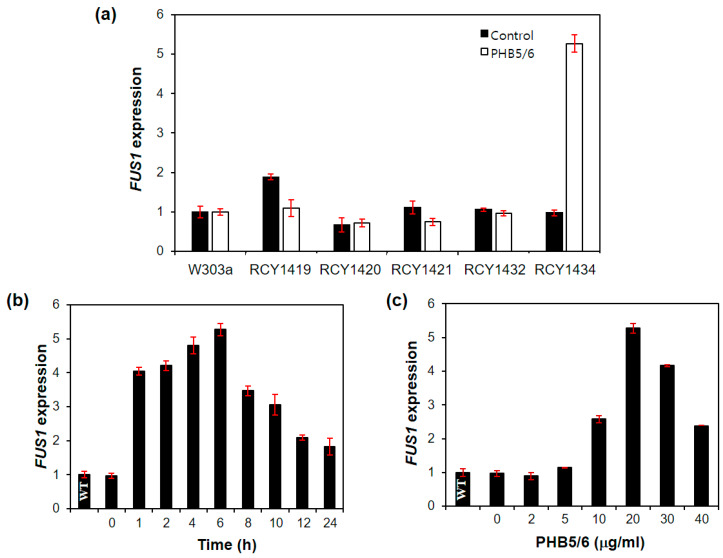
Construction of yeast strains for the study of *Lentinula edodes* and the mating pheromones (*PHB*)–pheromone receptors (*RCB*) interaction. (**a**) Modification of *S. cerevisiae* W303a. The expression of the *FUS1* gene was monitored by real-time PCR to measure the mating pheromone signal activation through the PHB–RCB interaction. PHB5 and PHB6 were added at a concentration of 20 μg/mL for 6 h. The mean difference in RCY1419 was not statistically significant (*p* > 0.05). (**b**) Optimal time for PHB treatment. (**c**) Optimal concentration of PHB for maximum gene expression. Basal level of the expressed *FUS1* in the actively growing wild-type strain without PHB treatment (the bar marked as “WT”) was used as an experimental control for (**b**,**c**). Both experiments were conducted at 30 °C. The *FUS1* expression relative to the β-tubulin gene expression was normalized using triplicated data. Error bars indicate standard deviations of means.

**Figure 2 genes-11-00506-f002:**
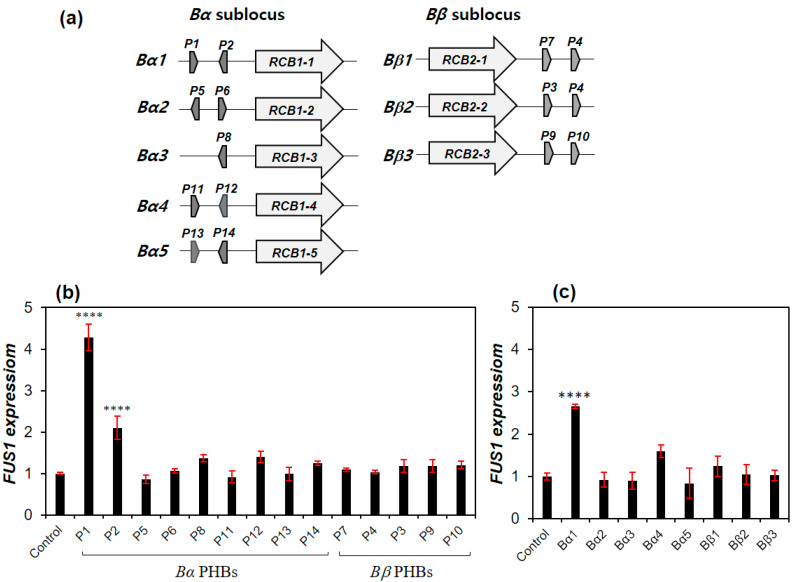
Interaction of *Lentinula edodes* mating pheromone receptors with mating pheromones. (**a**) Structure of the *B* mating-type locus in *L. edodes*. Allelic variants of *RCB1* and *RCB2* for *Bα* and *Bβ* subloci, respectively, are composed of 15 different combinations of the *B* mating-type locus. The picture is reproduced from our previous study [27]. (**b**) Interaction of RCB1–2 with different mating pheromones. P1–P14 represent the mating pheromones, PHB1–PHB14. Individual PHB at a concentration of 20 μg/mL was added to the yeast RCY1433 strain at 30 °C for 6 h. (**c**) Treatment of PHB pairs representing each sublocus shown in (**a**). PHBs in total 20 μg/mL (10 μg/mL of each PHB) were added to RCY1433 at 30 °C for 6 h. PHB8, sole PHB representing the *Bα3* sublocus, was added as 20 μg/mL. Basal level of the expressed *FUS1* in RCY1433 without PHB treatment was used as an experimental control. The *FUS1* expression relative to the β-tubulin gene expression was normalized using triplicated data. Error bars indicate standard deviations of means. The statistical significance of the mean difference between PHB-treated sample and the untreated sample (control) is indicated on the bar with asterisks (**** for *p* ≤ 0.0001, in this case).

**Figure 3 genes-11-00506-f003:**
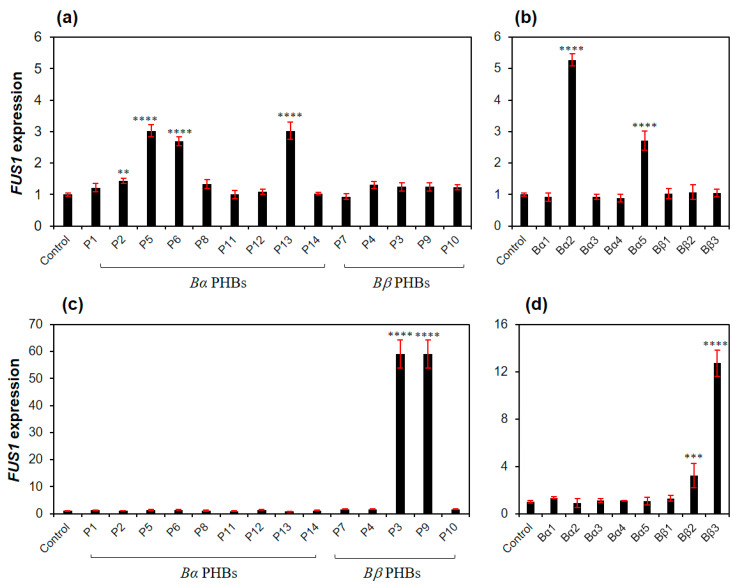
Interactions of RCB1-4 and RCB2-1 with PHBs. (**a**,**b**) interaction of RCB1-4 with single PHBs (**a**) and PHB pairs (**b**–**d**) interaction of RCB2–1 with single PHBs (**c**) and PHB pairs (**d**) a concentration of 20 μg/mL of individual PHB or PHBs in a pair was added to RCY1434 or RCY1440 at 30 °C for 6 h. Basal level of the expressed *FUS1* in RCY1434 or RCY1440 without PHB treatment was used as an experimental control. The *FUS1* expression relative to the β-tubulin gene expression was normalized using triplicated data. Error bars indicate standard deviations of means. The statistical significance of the mean difference between PHB-treated sample and the untreated sample (control) is indicated on the bar with asterisks (** for *p* ≤ 0.01, *** for *p* ≤ 0.01 and **** for *p* ≤ 0.0001).

**Figure 4 genes-11-00506-f004:**
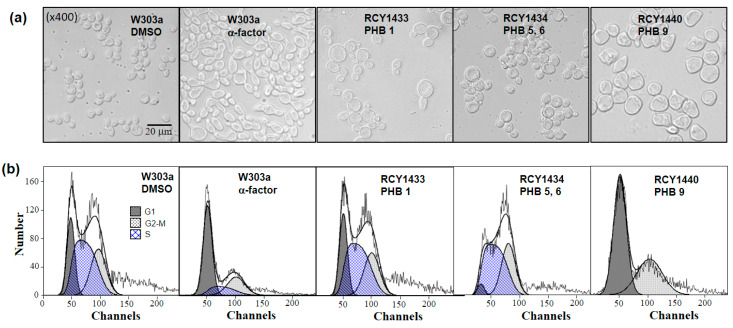
Effects of the PHB–RCB interaction on cell morphology and cell division cycle. (**a**) The yeast cell morphology 6 h after treatment of PHBs at a concentration of 20 μg/mL. (**b**) Analysis of cell division cycle using flow cytometry.

**Figure 5 genes-11-00506-f005:**
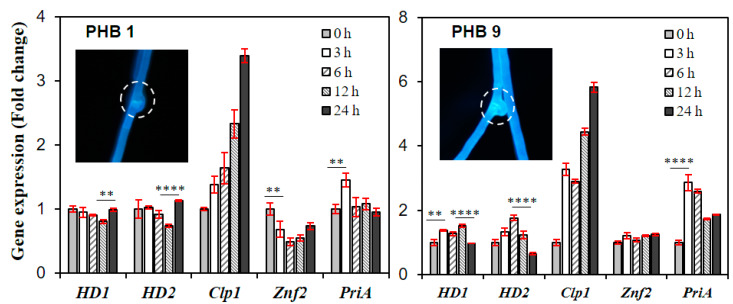
Effect of PHBs on the monokaryotic S1–10 strain of *Lentinula edodes*. Expression of the downstream genes in the mating pathway was monitored time-dependently. PHB1 or PHB9 (20 μg/mL) was added to actively growing mycelia of *L. edodes* S1–10 strain at 25 °C. Inset pictures show the microscopic image of the mushroom mycelia after the treatment of PHB1 or PHB9. The mycelia were stained with calcofluor. The circles indicate clamp connections. Error bars indicate standard deviations of means. The statistical significance of the mean difference is indicated on the bar with asterisks (** for *p* ≤ 0.01, and **** for *p* ≤ 0.0001).

**Table 1 genes-11-00506-t001:** Yeast strains used in this study.

Strain Name	Genotype
W303a	*MAT-a ade2-1 ura3-1 his3-11, 15 trp1-1 leu2-3, 112 can1-100*
RCY1419	W303a *ste2Δ*::*KanMX6^RCB1^*^-4^
RCY1420	W303a *sst2Δ*::*HIS3*
RCY1421	W303a *sst2Δ*::*HIS3*, *ste2Δ::KanMX6^RCB1-4^*
RCY1423	W303a *sst2Δ*::*HIS3*, *ste2Δ::KanMX6^RCB2-1^*
RCY1429	W303a *sst2Δ*::*HIS3*, *ste2Δ::KanMX6^RCB1-2^*
RCY1432	W303a *sst2Δ*::*HIS3*, *gpa_Le_*::*TRP1*
RCY1433	W303a *sst2Δ*::*HIS3*, *gpa_Le_*::*TRP1*, *ste2Δ::KanMX6^RCB1-2^*
RCY1434	W303a *sst2Δ*::*HIS3*, *gpa_Le_*::*TRP1*, *ste2Δ::KanMX6^RCB1-4^*
RCY1440	W303a *sst2Δ*::*HIS3*, *gpa_Le_*::*TRP1*, *ste2Δ::KanMX6^RCB2-1^*

**Table 2 genes-11-00506-t002:** Sequence of synthetic pheromones.

Sublocus	PHBs	Sequence	Sublocus	PHBs	Sequence
***Bα1***	PHB1	EHDTADSTNIGYAC-OME *	***Bβ1***	PHB7	EAIGAGDATAFC-OME
	PHB2	EHDTSDSGYTGYC-OME		PHB4	EAGGGDAIAFC-OME
***Bα2***	PHB5	EHPSDSGAVADFGYC-OME	***Bβ2***	PHB3	EAVGSGDIIGFC-OME
	PHB6	EHADESGSVALLGGYC-OME		PHB4	EAGGGDAIAFC-OME
***Bα3***	PHB8	EHDTNDSAFLGFC-OME	***Bβ3***	PHB9	EAVGSGDIIGFC-OME
***Bα4***	PHB11	ERPSNSGAVADFGYC-OME		PHB10	EAIGAADGSAFC-OME
	PHB12	EHDSSDSTDIGYC-OME			
***Bα5***	PHB13	EHPSETSSDANFGSYC-OME			
	PHB14	EHDTSDSTDIGYC-OME			

* OME indicates the carboxymethylation at the C-terminal cysteine residue.

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
