# Peer review of "Investigation of Mating Pheromone–Pheromone Receptor Specificity in Lentinula edodes"

_genes, 2020, doi:10.3390/genes11050506_

Round 1

Reviewer 1 Report

Manuscript genes-786623

The manuscript by Kim et al. describes an analysis of compatible pheromones and pheromone receptor genes from different alleles of the B mating type locus of the basidiomycete Lentinula edodes. This mushroom-forming fungus contains a complex B locus, which is subdivided into two subloci (B-alpha and B-beta), each of which can occur in several possible alleles found in different strains in nature. The authors used a yeast system where the yeast STE2 gene was substituted by an L. edodes pheromone receptor gene, and the yeast G-alpha gene Gpa1 was modified for better compatibility of the corresponding G-protein with L. edodes receptor proteins. The yeast strains carrying L. edodes receptor genes were treated with synthetic pheromones (without the farnesylation that is predicted to occur in nature). Several compatible interactions were identified by expression analysis of genes activated by the downstream MAPK cascade and by changes in yeast morphology. Selected pheromones were also tested for activity on a monokaryotic L. edodes strain by testing for expression of genes expected to be differentially upregulated during mating as well as by checking for clamp connections, which are a sign of successful mating and formation of dikaryotic strains. The results are of great interest to researchers working on sexual development in fungi and researchers interested in the function of self/non-self recognition systems in general.

There are just a few points where the manuscript could be improved:

  1. Figure S5 was not included in the material for download.
  2. Please explain why in the yeast strains that were used for testing, STE2 was substituted by the L. edodes receptor genes and not STE3, which would be the homolog of the L. edodes receptor.
  3. Table 2: The asterisk for the footnote in Table 2 appears only to relate tot he PHB1 pheromone (where it is placed at the end, not the actual cysteine). Does this mean that the Carboymethylation was only present in the synthetic PHB1 pheromone?
  4. Figures 1 and 2: In sections b) and c), it should be explained what was used as control (strains, time points and/or concentrations of treatment etc.).
  5. In line 182, it is stated that not all pheromone/receptor combinations worked in the yeast system. Is there anything in the sequences of the pheromones/receptors that might indicate that certain combinations might work better? It would be helpful to include alignments of pheromones and receptors in the supplement, and perhaps include hypotheses as to why some combinations work better than others in the discussion.
  6. Line 191: The sentence "... including PHB5 and PHB5, and the self PHBs...." is somewhat confusing. PHB5 would actually be a self PHB for receptor RCB1-2.
  7. Figure 3: Please explain what was used as controls.
  8. Figure 5 and methods section: It should be explained how the tests were conducted for applying the synthetic pheromones to L. edodes. What were the culture conditions (solid or liquid medium, what type of medium, inoculum, time until application of pheromone, how was the pheromone applied and in what concentration etc.).

Author Response

The manuscript by Kim et al. describes an analysis of compatible pheromones and pheromone receptor genes from different alleles of the B mating type locus of the basidiomycete Lentinula edodes. This mushroom-forming fungus contains a complex B locus, which is subdivided into two subloci (B-alpha and B-beta), each of which can occur in several possible alleles found in different strains in nature. The authors used a yeast system where the yeast STE2 gene was substituted by an L. edodes pheromone receptor gene, and the yeast G-alpha gene Gpa1 was modified for better compatibility of the corresponding G-protein with L. edodes receptor proteins. The yeast strains carrying L. edodes receptor genes were treated with synthetic pheromones (without the farnesylation that is predicted to occur in nature). Several compatible interactions were identified by expression analysis of genes activated by the downstream MAPK cascade and by changes in yeast morphology. Selected pheromones were also tested for activity on a monokaryotic L. edodes strain by testing for expression of genes expected to be differentially upregulated during mating as well as by checking for clamp connections, which are a sign of successful mating and formation of dikaryotic strains. The results are of great interest to researchers working on sexual development in fungi and researchers interested in the function of self/non-self recognition systems in general.

There are just a few points where the manuscript could be improved:

  1. Figure S5 was not included in the material for download.

Response: I forgot to include Figure S5 while reformatting the manuscript upon responding editor’s request. Figure S5 is now included.

  1. Please explain why in the yeast strains that were used for testing, STE2 was substituted by the L. edodes receptor genes and not STE3, which would be the homolog of the L. edodes receptor.

Response: We chose STE2 to compare the yeast alpha-factor-STE2 interaction with the L.edodes PHB-RCB interaction in the yeast model. This is because we were not able to obtain farnesylated synthetic PHBs.

  1. Table 2: The asterisk for the footnote in Table 2 appears only to relate tot he PHB1 pheromone (where it is placed at the end, not the actual cysteine). Does this mean that the Carboymethylation was only present in the synthetic PHB1 pheromone?

Response: All PHBs have the carboxymethylation. We changed the footnote as “*OME indicates the carboxymethylation at the C-terminal cysteine residue.”

  1. Figures 1 and 2: In sections b) and c), it should be explained what was used as control (strains, time points and/or concentrations of treatment etc.).

Response: We used the basal level of the expressed FUS1 in the wild-type strain without PHB treatment as an experimental control. The basal FUS1 level was coincident with the RCB carrying yeast at 0 time with PHB treatment (b) or no PHB treatment (c). We included following description at the figure legend: “Basal level of the expressed FUS1 in the actively growing wild-type strain without PHB treatment (the bar marked as “WT”) was used as an experimental control for b) and c).”

  1. In line 182, it is stated that not all pheromone/receptor combinations worked in the yeast system. Is there anything in the sequences of the pheromones/receptors that might indicate that certain combinations might work better? It would be helpful to include alignments of pheromones and receptors in the supplement, and perhaps include hypotheses as to why some combinations work better than others in the discussion.

Response: We were not able to draw any plausible explanation on this because there can be many factors involved in the PHB-RCB interactions, including 3D-folding of synthetic PHBs, access of the PHBs to the RCB on the surface of yeast cell, and the absence of C-term farnesylation. The sequence comparisons were published in our previous paper [27]. We discussed this in line 319-325. We probably need better model that mimics the farnesylated PHBs.

  1. Line 191: The sentence "... including PHB5 and PHB5, and the self PHBs...." is somewhat confusing. PHB5 would actually be a self PHB for receptor RCB1-2.

Response: We made mistake on this. We corrected this as follows:

“In the single PHB treatment, only PHB1 and PHB2 showed a 4.3-fold and 2.1-fold increase in the FUS1 expression, respectively, while there was no response by other PHBs, including the self PHBs (PHB5 and PHB6) (Figure 2b).”

  1. Figure 3: Please explain what was used as controls.

Response: Control experiment was conducted using the wild-type strain without PHB treatment. We added following sentence in the figure legends (Figures 2 and 3).

For Figure 2: “Basal level of the expressed FUS1 in RCY1433 without PHB treatment was used as an experimental control.”

For Figure 3: “Basal level of the expressed FUS1 in RCY1434 or RCY1440 without PHB treatment was used as an experimental control.”

  1. Figure 5 and methods section: It should be explained how the tests were conducted for applying the synthetic pheromones to L. edodes. What were the culture conditions (solid or liquid medium, what type of medium, inoculum, time until application of pheromone, how was the pheromone applied and in what concentration etc.).

Response: Following experimental procedure were added in the Materials and methods section 2.6.

“2.6. Effect of synthetic PHBs on Lentinula edodes S1-10

      The monokaryotic S1-10 strain of Lentinula edodes was grown in PDB (100 mL) for a week at 25°C. The mycelia were harvested by centrifugation. The collected mycelia were suspended in fresh PDB and were fragmented by shaking (150 rpm) in the presence of 5 pieces of glass bead (d = 1 cm, Sigmund Linder GmbH, Germany) for 12 h. The culture broth was incubated additional 2 days after removal of the glass beads. The mycelia were washed twice with fresh PDB after centrifugation. The mycelia were resuspended in 100 mL PDB. Synthetic PHB (final 20 mg/mL) was treated to the mycelial suspension at 25°C. Samples (10 mL) were taken at given time intervals and were subjected to total RNA extraction. The real-time qRT-PCR analysis was performed using the same procedure described above with primer sets specific to target genes (Table S2).

      For the microscopic analysis, the PHB-treated mycelia were stained with equivalent mixture of 10% potassium hydroxide and calcofluor white (Sigma-Aldrich), which binds to the chitin component of the fungal cell wall. The stained mycelia were examined under fluorescence microscope (DP-70, Olympus, Japan) at 1000X magnification with excitation at 355 nm.”

Reviewer 2 Report

This manuscript by Kim et al. addresses the specificity of mating pheromones (PHBs) and mating pheromone receptors (RCBs) in Lentinula edodes, one of the mushrooms. Fungi belonging to Basidiomycota have a similar mating pathway via the recognition of mating pheromones by their receptor like Ascomycete yeasts. The mycelial monokaryotic hyphae fuse to generate a dikaryotic hyphae. This mating process is primarily required for the compatibility between PHBs and RCBs that play an important role in non-self recognition. In L. edodes, B mating-type locus is highly complex (like all Basidiomycota). It contains five subloci with nine PHBs and five RCBs, and three subloci with five PHBs and three RCBs. So far, the specificity between the 14 pheromones and the 8 receptors has not been verified yet. In this study, they used Saccharomyces cerevisiae model system that express L. edodes receptors instead of its mating pheromone receptor STE3 to investigate the specificity between PHBs and RCBs. By monitoring the expression of FUS1, a responsible gene for mating, in S. cerevisiae a-cells, they examined whether C-terminally carboxymethylated synthetic pheromone peptides (without farnesylation) are recognized by receptors. Thus, they found some compatible combinations of PHBs and RCBs (e.g., PHB1,2 and RCB1-2). The PHB–RCB interaction is also confirmed by showing the elevated expression of clp1, which involved in the formation of clamp cells, considered as a hallmark of successful mating in most Basidiomycota.

In my opinion, this is an important study, and the experiments are often appropriately controlled. However, I think that the interpretation and presentation of data need to be improved according to the suggestions.

General comments:

  1. It is not clear to me why the authors used the W303a strain (‘a’-cells of S. cerevisiae). Mushroom pheromone receptors are homologous to STE3 (the receptor for a-factor) of S. cerevisiae (Lines 48–50), therefore I think that the expression of receptors should be induced in ‘α’-cells. Even if the pheromone signaling pathway downstream of the activated receptors is shared in both cell types, promoters for receptor are often the opposite mating-type pheromone-inducible in yeasts. I have serious concerns about two things: 1) the expression level of L. edodes receptors in the W303a strain is enough, and 2) the signal is normally transmittedfrom the cell-surface receptor.
  2. The authors used synthetic pheromone peptide with C-terminal carboxymethylation without farnesylation. In ref. 12, this type of peptides remains active in the mating pathway activation in L. edodes (Lines 78–79), but farnesylation of pheromone peptides is a requirement for their biological activity in almost all yeasts. Why not do the authors use farnesylated peptides? Therefore, I think that the interpretation and presentation of data should be interpreted more cautiously. For example, Fig. S4 show the effects of α-factor on the expression of FUS1 in a-cells. Mating pheromones of mushrooms are lipid-peptides, so it would better investigated by using a-factor without farnesylation as a control.
  3. In Fig. 3, the treatment of two kinds of pheromones, Bβ2 (PHB3 and PHB4) and Bβ3 (PHB9 and PHB10), is not higher than the individual treatment of PHB3 of PHB9. Is this due to low concentration of peptides (10 µg/ml)? Treatment of these PHBs may show anergic effects.
  4. In the current version, the authors does not fully explain about relationship between amino acid sequences of pheromones/receptor and their compatibility. I would appreciate if a clearer explanation would be provided.

Minor comments:

  1. Fig S5 file is not available.
  2. A sentence (, while there was … the self PHBs (Figure 2b); Lines 190–191) should be rephrased as clearly as possible.

Author Response

Dear reviewer, 

I'd like to thank for your kind suggestions and comments. It helps greatly to improve our manuscript. We tried hard to respond to your comments. We also corrected the errors in English through the MDPI English correction services. We hope the revised version suffice your concerns.

Best wishes, 

HS Ro, on behalf of authors

Reviewers' comments and responses.

This manuscript by Kim et al. addresses the specificity of mating pheromones (PHBs) and mating pheromone receptors (RCBs) in Lentinula edodes, one of the mushrooms. Fungi belonging to Basidiomycota have a similar mating pathway via the recognition of mating pheromones by their receptor like Ascomycete yeasts. The mycelial monokaryotic hyphae fuse to generate a dikaryotic hyphae. This mating process is primarily required for the compatibility between PHBs and RCBs that play an important role in non-self recognition. In L. edodesB mating-type locus is highly complex (like all Basidiomycota). It contains five  subloci with nine PHBs and five RCBs, and three  subloci with five PHBs and three RCBs. So far, the specificity between the 14 pheromones and the 8 receptors has not been verified yet. In this study, they used Saccharomyces cerevisiae model system that express L. edodes receptors instead of its mating pheromone receptor STE3 to investigate the specificity between PHBs and RCBs. By monitoring the expression of FUS1, a responsible gene for mating, in S. cerevisiae a-cells, they examined whether C-terminally carboxymethylated synthetic pheromone peptides (without farnesylation) are recognized by receptors. Thus, they found some compatible combinations of PHBs and RCBs (e.g., PHB1,2 and RCB1-2). The PHB–RCB interaction is also confirmed by showing the elevated expression of clp1, which involved in the formation of clamp cells, considered as a hallmark of successful mating in most Basidiomycota.

In my opinion, this is an important study, and the experiments are often appropriately controlled. However, I think that the interpretation and presentation of data need to be improved according to the suggestions.

General comments:

  1. It is not clear to me why the authors used the W303a strain (‘a’-cells of  cerevisiae). Mushroom pheromone receptors are homologous to STE3 (the receptor for a-factor) of S. cerevisiae(Lines 48–50), therefore I think that the expression of receptors should be induced in ‘α’-cells. Even if the pheromone signaling pathway downstream of the activated receptors is shared in both cell types, promoters for receptor are often the opposite mating-type pheromone-inducible in yeasts. I have serious concerns about two things: 1) the expression level of L. edodes receptors in the W303a strain is enough, and 2) the signal is normally transmitted from the cell-surface receptor.

Response:

First, we chose STE2 to compare the yeast alpha-factor-STE2 interaction with the L.edodes PHB-RCB interaction in the yeast model, because we were only able to acquire soluble synthetic PHBs. We don’t think choosing a-strain instead of alpha-strain make big difference since we just utilize the yeast as a reporting system for RCB-PHB interaction. Second, all RCB were integrated into the yeast chromosome replacing STE2, thus were under the control of the endogenous STE2 promoter. We checked the expression of RCBs in yeast by RT-PCR and the expression levels were equivalent to the STE2 expression. Third, we have checked the transmission of signal through the expression of FUS1 gene which is a downstream component of yeast mating pheromone signal transduction pathway. Therefore, we believe that the mating pheromone signal produced from the RCB-PHB interaction is normally transmitted to the yeast mating pathway.

  1. The authors used synthetic pheromone peptide with C-terminal carboxymethylation without farnesylation. In ref. 12, this type of peptides remains active in the mating pathway activation in  edodes(Lines 78–79), but farnesylation of pheromone peptides is a requirement for their biological activity in almost all yeasts. Why not do the authors use farnesylated peptides? Therefore, I think that the interpretation and presentation of data should be interpreted more cautiously. For example, Fig. S4 show the effects of α-factor on the expression of FUS1 in a-cells. Mating pheromones of mushrooms are lipid-peptides, so it would better investigated by using a-factor without farnesylation as a control.

Response: We totally agree with you. However, we could not obtain the farnesylated peptides from commercial synthetic services and that is why we have focused on few selected PHB-RCB interactions which have shown distinct signals. We’d like to ask the reviewer’s generous understanding on this.  

  1. In Fig. 3, the treatment of two kinds of pheromones, Bβ2 (PHB3 and PHB4) and Bβ3 (PHB9 and PHB10), is not higher than the individual treatment of PHB3 of PHB Is this due to low concentration of peptides (10 µg/ml)? Treatment of these PHBs may show anergic effects.

Response: We think the reduction of FUS1 expression by PHB pair may be due to the crowding effect that restricts the access of PHB3 or PHB9 to RCB2-1. It is also possible that there is uncharacterized interaction between the pair of PHBs so to reduce the effective concentration of PHB3 or PHB9. For the reflection of this, we modified line 232 in the main text as follows: “However, the treatment of both pheromones from Bb2 (PHB3 and PHB4) or Bb3 (PHB9 and PHB10) was rather inferior to the individual treatment of PHB3 or PHB9 (Figure 3d), suggesting that the presence of nonspecific pheromone inhibits the specific PHBRCB interaction.”

  1. In the current version, the authors does not fully explain about relationship between amino acid sequences of pheromones/receptor and their compatibility. I would appreciate if a clearer explanation would be provided.

Response: The differential features in RCBs and PHBs were described in our previous paper [27]. We added following new descriptions in Discussion to explain the specificity of RCB-PHB interactions.

“ It appears that RCBs in L. edodes have a specificity toward a certain group of PHBs. In our previous study, we categorized the L. edodes PHBs into three different groups based on the amino acid composition: PHB1, PHB2, PHB8, and PHB14 to group I; PHB5, PHB6, PHB11, and PHB13 to group II; and PHB3, PHB9, PHB7, and PHB10 to group III [27]. RCB1-2 in this study shows interaction with the group I PHBs (PHB1 and PHB2) whereas RCB1-4 and RCB2-1 interact with the group II PHBs (PHB5, PHB6, and PHB13) and group III PHBs (PHB3 and PHB9), respectively. The results imply that the membrane-embedded RCBs have group-specific recognition motifs in their loop domains, potentially in the L3, L5, and L7 domains described in our previous report [27]. It also implies that the nonself recognition in the mushroom mating can occur though the PHB group-specific interaction with RCBs.”

Minor comments:

  1. Fig S5 file is not available.

Response: Fig S5 is included. It was newly included during editors precheck but not transferred to the reviewer.

  1. A sentence (, while there was … the self PHBs (Figure 2b); Lines 190–191) should be rephrased as clearly as possible.

Response: Errors were corrected as follows: “In the single PHB treatment, only PHB1 and PHB2 showed a 4.3-fold and 2.1-fold increase in the FUS1 expression, respectively, while there was no response by other PHBs, including the self PHBs (PHB5 and PHB6) (Figure 2b).”
